# Dry Reforming of Propane over γ-Al₂O₃ and Nickel Foam Supported Novel SrNiO₃ Perovskite Catalyst

**Sudhakaran M.S.P, Md. Mokter Hossain, Gnanaselvan Gnanasekaran and Young Sun Mok ***

Department of Chemical and Biological Engineering, Jeju National University, Jeju 63243, Korea;
mspsudhakaran@gmail.com (S.M.S.P.); mokterm@gmail.com (M.M.H.); gsel1033@gmail.com (G.G.)
* Correspondence: smokie@jejunu.ac.kr; Tel.: +82-064-754-3682; Fax: +82-64-755-3670

**Abstract:** The SrNiO₃ perovskite catalyst was synthesized by the citrate sol-gel method and supported on γ-Al₂O₃ and Nickel foam, which was used to produce syngas (CO and H₂) via dry reforming of propane (DRP). Several techniques characterized the physicochemical properties of the fresh and spent perovskite catalyst. The X-ray diffractograms (XRD) characterization confirmed the formation of the perovskite compound. Before the catalytic activity test, SrNiO₃ perovskite catalyst was reduced in the H₂ atmosphere. Results indicated that the H₂ reduction slightly increased the activity of the SrNiO₃ perovskite catalyst. The catalytic activity was examined for the $CO_2/C_3H_8$ ratio of 3 and reaction temperatures in the range of 550 °C–700 °C. The results from the catalytic study achieved 88% conversion of $C_3H_8$ and 66% conversion of $CO_2$ with SrNiO₃/NiF at 700 °C. Also, syngas with a maximum concentration of 21 vol.% of CO and 29 vol.% of H₂ was produced from the DRP. The strong basicity of SrNiO₃ perovskite enhanced the CO selectivity, resulting in minimal carbon formation. Post reaction catalyst characterization showed the presence of carbon deposition which could have originated from propane decomposition.

**Keywords:** alkali earth metal; SrNiO₃; perovskite; propane; syngas; nickel foam; γ-Al₂O₃

---

## 1. Introduction

The conventional production of synthesis gas (syngas) by methane steam reforming (1) regularly produces a product with higher H₂/CO values greater than 3 [1,2]. Recently, the attention has been drawn to the conversion of light hydrocarbons with carbon dioxide into the valuable product (syngas) by catalytic reactions, which is known as dry reforming (2) [3]. The result of a product is with a lower H₂/CO ratio ≤2. This ratio is more suitable for the synthesis of liquid fuels, and chemicals such as olefins, methanol synthesis and Fisher–Tropsch synthesis [4–8]. Propane is a by-product of natural gas and majorly produced in a variety of petroleum refining operations. Primarily, it readily activates at a lower reaction temperature than methane [9].

$$CH_4 + H_2O = CO + 3H_2 \left( \Delta H^{\circ}_{298} = 222.4 \text{ kJmol}^{-1} \right) \tag{1}$$

$$C_3H_8 + 3CO_2 = 6CO + 4H_2 \left( \Delta H^{\circ}_{298} = 644.1 \text{ kJmol}^{-1} \right) \tag{2}$$

However, the steam and dry are an endothermic reaction, which requires high energy to occur reaction. Numerous catalysts have been described for each of the processes mentioned above for various hydrocarbons [10–12]. Noble metal catalysts show superior performance regarding the activity and the durability than non-noble metal catalysts. The Ni-based catalysts are attractive and promising due to their high activity, low cost, and abundant availability [13]. It has been widely studied with different support materials such as Al₂O₃, SiO₂, CeO₂, and SiC, and among them, Al₂O₃ is the most

widely used as catalytic support in dry reforming [14,15]. However, the main limitations of Ni catalyst are particle sintering and coke formation. Therefore, the development of a suitable catalyst with high activity and anti-coking capability is essential for dry reforming and the selection of suitable support may be a pivotal step to prepare the catalyst with high performance. The typical perovskites are $ABO_3$ type crystal structured materials with rare earth or alkaline earth metal at the A-site and a transition metal at the B-site [16,17]. These possess some exciting chemical and physical properties, e.g., high thermal stability, the excellent reactivity of lattice oxygen, low cost and abundant resources [16]. A-site replacement with alkaline earth metal is expected to enhance the carbon susceptible and thermal stability, and the B-site alteration is expected to increase the activity of the catalyst [18,19]. In last decade, many researchers have been widely reported $LaNiO_3$ [20–22], $LaFeO_3$ [23], $SmCoO_3$ [24] and also partial substitution of B- site perovskite catalysts show the higher activity, stability, and resistance to coke formation even at elevated temperature. The perovskite catalysts increase the syngas production in dry reforming reaction [25–27]. Metal foams are highly porous with open-celled materials, which pack in a three-dimensional network. Metal foams have promising support due to their low density, mechanical strength, and high thermal conductivity. It has been reported in heterogeneous catalysis include dry reforming and catalytic combustion of $CH_4$ [28,29].

Hence, this study has focused on low-cost and high basicity material with key features such as high catalytic activity and coke resistance. The objectives of this study are to synthesize of novel $SrNiO_3$ perovskite as a catalyst to produce syngas via dry reforming of propane. The physicochemical properties of catalysts characterized with various techniques and the catalytic activity investigated in typical experimental gas reactor setup. The catalytic conversion of reactants and enhancement of production of syngas was carried out with the effects of reaction temperature. The effects of support materials were examined with $SrNiO_3$ perovskite by a different form of support materials such as $\gamma$-$Al_2O_3$ and nickel foam. Also, the effect of $H_2$ reduction was studied by comparing the conversion of the reduced and unreduced $SrNiO_3$ perovskite catalyst before the propane dry reforming reaction.

## 2. Result and Discussion

### 2.1. Materials Characterization

XRD patterns of the calcined perovskite and reduced perovskite are shown in Figure 1. The sample was calcined at 900 °C and reduced by hydrogen at 700 °C. The fresh catalyst shows the real pattern of $SrNiO_3$ perovskite state and the somewhat slight secondary peak of NiO also found. As seen in Figure 1, the highest sharp peaks found at 2θ with a value of 32.52°. It shows the formation of the $SrNiO_3$ crystalline phase. The other peaks at 2θ values of 19.24°, 29.36°, 39.04°, 41.08°, 44.04°, 49.84°, 56.24°, 58.24°, 68.4°, and 76.24° also correspond to $SrNiO_3$ species. Beside $SrNiO_3$ phase, some peaks were evolved at 37.04°, 43.08°, and 62.76°, which belong to the NiO crystalline phase. All Ni precursors were not used to form $SrNiO_3$. The desired $SrNiO_3$ catalyst was synthesis successfully, and the diffraction pattern confirms formation of the hexagonal perovskite structure, matched with standard pattern (JCPDS card no: 33-1347) [30]. The post-reduction catalyst clearly shows the diffraction peaks corresponding to $Ni^0$ and SrO. The peaks found at 2θ values of 44.4°, 51.8°, and 76.3° correspond to metallic Ni and peaks were evolved at 31.4°, 36.3°, and 62.2° correspond to the SrO crystalline phase. Those peaks were matched with the standard patterns (JCPDS card no: 65-2865 and JCPDS card no: 27-1304), respectively [31,32]. Generally, the high-temperature reduction of perovskites leads to the formation of nanoparticles of B site metal (e.g., Ni) dispersed on the oxide formed by the A site metal (e.g., Sr) [33,34]. The XRD diffraction of $\gamma$-$Al_2O_3$ and NiF supported catalysts are shown in Figure 1b,c. The results indicate the existence of $SrNiO_3$, NiO, Ni, SrO peaks in both supported catalysts. The peaks were observed at 2θ values of 45.3°, and 66.7° belong to the $\gamma$-$Al_2O_3$ (JCPDS card no: 02-1420) [35] crystalline phase and peaks were found at 2θ values of 44.4°, 51.8°, and 76.3° correspond to metallic Ni [36].

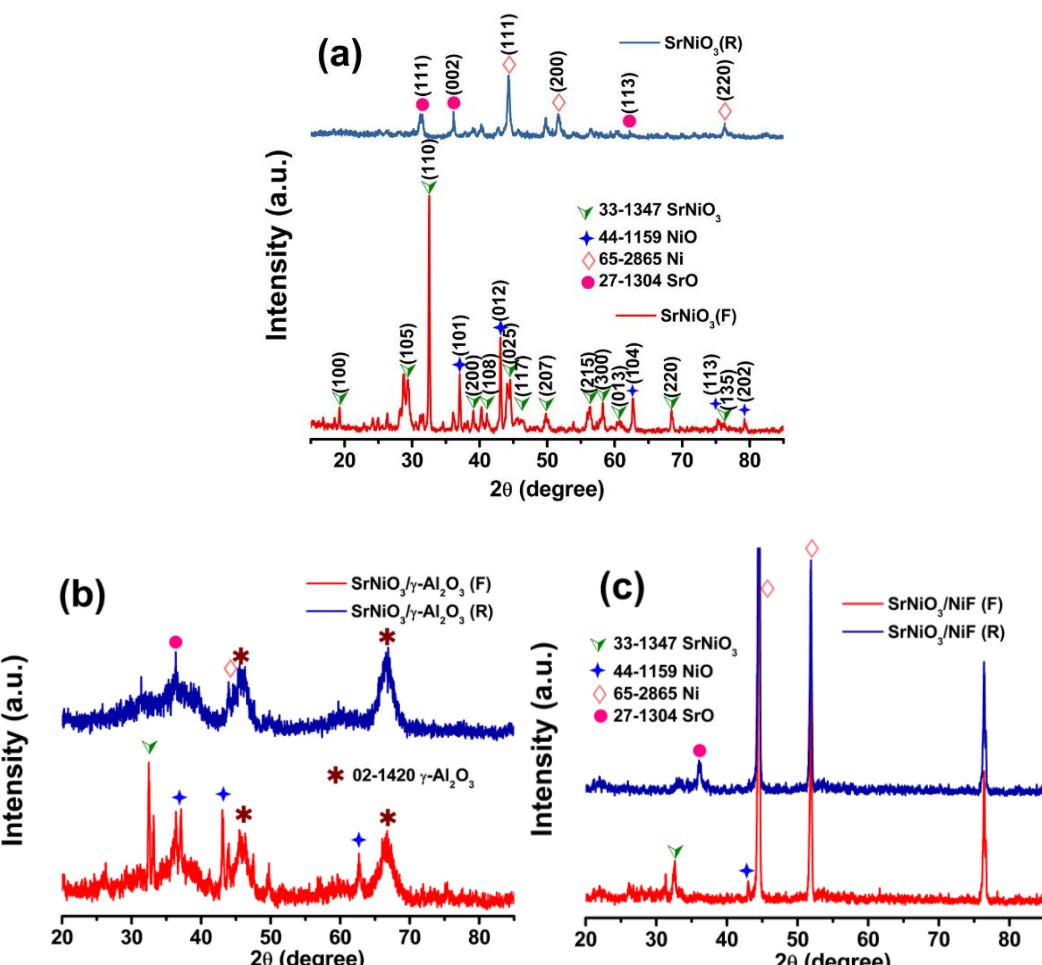

**Figure 1.** X-ray diffraction patterns of (**a**) SrNiO$_3$ unreduced (F) and reduced (R) perovskite catalysts (**b**) SrNiO$_3$ (F and R) supported in γ-Al$_2$O$_3$ (**c**) supported in nickel foam.

The N$_2$ physisorption isotherm of the SrNiO$_3$ perovskite is shown in Figure 2. The Brunauer–Emmett–Teller (BET) surface area is 3.3 m$^2$ g$^{-1}$, which is agreed with perovskite materials as reported in the literature [34]. The BET surface area of the catalysts with different supports reported in Table 1, the surface area of reduced catalyst was increased in both support materials compared to the fresh catalyst SrNiO$_3$. It implies that the breakdown of the perovskite structure, leading to formation of Ni/SrO, resulted in the generation of porosity to some extent, which also explains the slight enhancement of surface area of this bulk perovskite after reduction. The surface morphology of the SrNiO$_3$ perovskite catalyst is shown in Figure 3. The high-resolution SEM image reveals the porous and flake structure of the material. From low resolution and high-resolution SEM images, it shows the less impact of particle agglomeration due to the citrate sol-gel method [37].

**Table 1.** Brunauer–Emmett–Teller (BET) surface area of the different supported catalysts.

| Catalyst | S$_{BET}$ (m$^2$/g) |
| --- | --- |
| SrNiO$_3$/γ-Al$_2$O$_3$(F) | 205.1 |
| SrNiO$_3$/γ-Al$_2$O$_3$(R) | 223.5 |
| SrNiO$_3$/NiF(F) | 6.2 |
| SrNiO$_3$/NiF(R) | 23.7 |

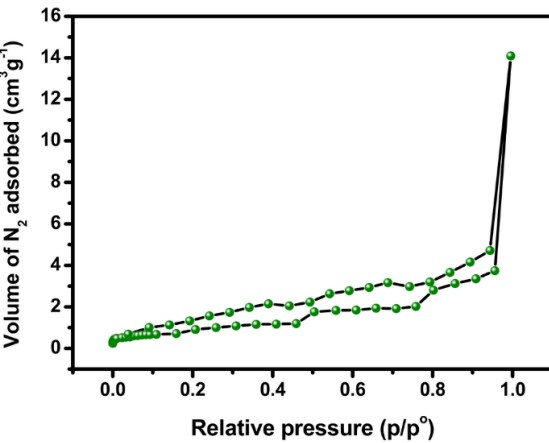

**Figure 2.** $N_2$-adsorption and desorption isotherm of $SrNiO_3$ perovskite catalyst.

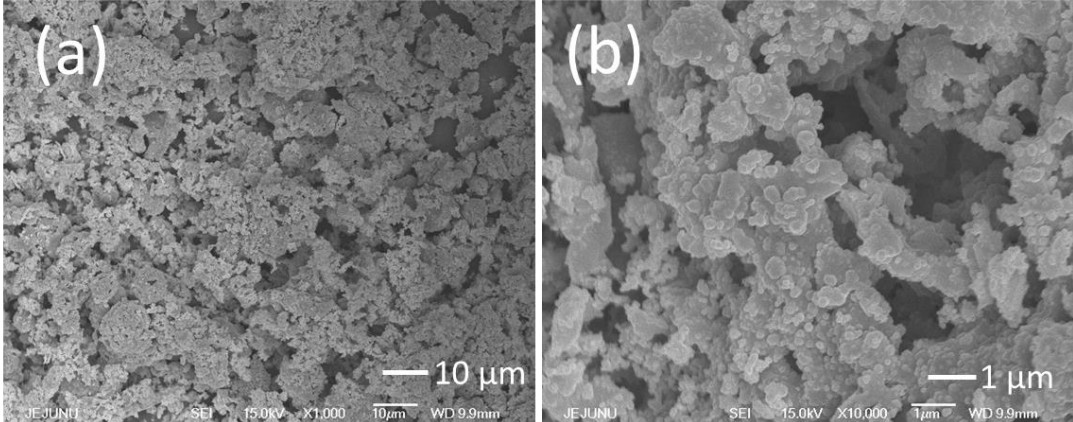

**Figure 3.** FE-SEM images of $SrNiO_3$ perovskite catalyst at low (**a**) and high magnifications (**b**).

Temperature programmed reduction ($H_2$-TPR) analysis was determined by the reduction behavior of the fresh catalyst and reduced catalyst. Figure 4 shows the $H_2$-TPR profile of the fresh catalyst and reduced $SrNiO_3$ perovskite catalyst. The two intense peaks were observed at lower and higher temperatures (600 K and 987 K, respectively) of the $SrNiO_3$ fresh catalyst. The reduced $SrNiO_3$ exhibited one broad peak at higher temperature 1096 K. The fresh calcined $SrNiO_3$ catalyst reduction process occurred by the following steps. The first step at lower temperature 600 K is attributed to $Ni^{3+}$ reduction to $Ni^{2+}$ and, the second peak at 987 K corresponds to the complete reduction of the perovskite to form $Ni^0$ (Equation (3)) [27].

$$SrNiO_3 + H_2 \rightarrow SrO + Ni^0 + 2H_2O \tag{3}$$

The reduced $SrNiO_3$ perovskite shows the single reduction peak at high temperature, which means the complete reduction of $SrNiO_3$ phase to SrO. The perovskite compound's reduction occurred at high temperature, which agreed with the previous report [38]. The reducibility of fresh and reduced $SrNiO_3$ perovskite supported with $\gamma$-$Al_2O_3$ and nickel foam is presented in Figure 4b. The reduction peaks were observed in the supported catalysts significantly matched with unsupported catalysts. The peaks were evolved at lower region 624 °K, 585 °K, and at higher region 948 °K, 970 °K of $\gamma$-$Al_2O_3$ and NiF supported catalysts confirmed the reduction properties of fresh $SrNiO_3$ perovskite. Those peaks were shifted to the lower temperature than an unsupported catalyst, which shows the interaction of support. The peak was found at around 1054 °K of both supported catalysts, which matched with the reduced $SrNiO_3$ perovskite. The reducibility of the both supported catalysts showed the metal support interaction enhance the accessibility of active species to the reactant.

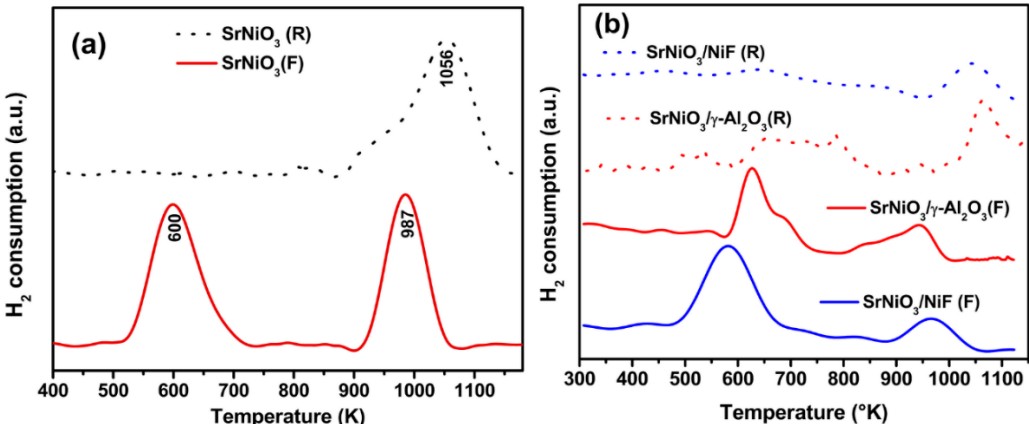

**Figure 4.** $H_2$-TPR of (**a**) unreduced (F) and reduced (R) SrNiO$_3$ perovskite unsupported catalysts and (**b**) SrNiO$_3$ (F and R) supported with γ-Al$_2$O$_3$, and nickel foam catalysts.

Figure 5 shows the chemisorption characters of SrNiO$_3$ perovskite by $H_2$ temperature programmed desorption method. As seen in Figure 5, the two distinct peaks were observed at 373 °C, and 426 °C. The first peaks attributed to the $H_2$ molecules desorbed from the metal particles, and they represent the outer positioned of Ni atoms in the perovskite lattice [39,40]. The second sharp peak attributed to the $H_2$ dissociated in the subsurface layers of the perovskite lattice. This phenomenon is known as $H_2$ spillover [41–43]. It reveals the high energy requires to unbind the $H_2$ molecules and strong enough of chemisorption. From the results, the amount of $H_2$-chemisorbed molecules was estimated under the curve. The $H_2$ consumption amount of the SrNiO$_3$ catalyst was $1.89 \times 10^{20}\ g_{cat}^{-1}$.

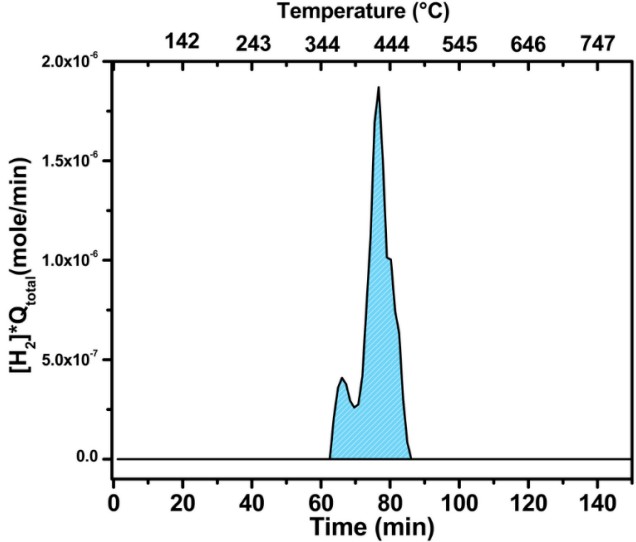

**Figure 5.** $H_2$-TPD of the SrNiO$_3$ perovskite catalyst.

## 2.2. Catalytic Activity

The activity of perovskite catalysts has been studied in two forms without reduction and reduced with $H_2$ before reaction. The reduced catalysts have shown the significant activity than a fresh catalyst. The reduction process happened to reduce the transition metal phase of the catalyst to a metallic phase, which is located at the active site of the catalyst [44]. It has been reported that reduction enhances the metal dispersion, thereby providing an adequate platform for propane dry reforming reaction to occur. Reduction step has been reported to improve the reactant conversion as well as syngas formation [45]. In this experiment, catalytic activity was examined to determine reactants

conversion for the reduced and unreduced catalyst with different supports. As seen in Figure 6, the catalytic activity of $SrNiO_3/\gamma-Al_2O_3$ (F and R) and $SrNiO_3/NiF$ (F and R) catalysts showed that increases conversion of $C_3H_8$ and $CO_2$ with increasing reaction temperature ranged from 550 °C to 700 °C. In terms of $C_3H_8$ conversion, the $SrNiO_3/\gamma-Al_2O_3$(R) showed that the higher conversion among all other catalysts. It showed a maximum 94% of $C_3H_8$ conversion at 700 °C and minimum 24% conversion at 550 °C, which is higher than the $SrNiO_3/\gamma-Al_2O_3$(F) catalyst. The $C_3H_8$ conversion of the Ni supported catalysts shows the increasing conversion with increases reaction temperature. The $C_3H_8$ conversion of $SrNiO_3/NiF$ (F and R) catalyst is 81% and 87% at 700 °C, respectively. The unreduced catalysts showed the lower $C_3H_8$ conversion with compare to the reduced catalysts. These indicated that in-situ catalyst reduction occurred by $H_2$ arising from the $C_3H_8$ cracking to produce $H_2$ and carbon [46].

Interestingly, the $CO_2$ conversion of the $SrNiO_3/NiF$ (F and R) catalysts showed the significant conversion of $CO_2$ than the $SrNiO_3/\gamma-Al_2O_3$ (F and R) catalyst. Among all the catalysts, $SrNiO_3/NiF$(R) showed the 69% of $CO_2$ conversion, which was 13% higher than the $SrNiO_3/\gamma-Al_2O_3$(R). It suggested that the $SrNiO_3/NiF$(R) leads to the dry reforming reaction even at increasing temperature while the $SrNiO_3/\gamma-Al_2O_3$ (F and R) has no significant ability to convert $CO_2$ to CO since it leads the $C_3H_8$ cracking reaction and produces more carbon on the surface.

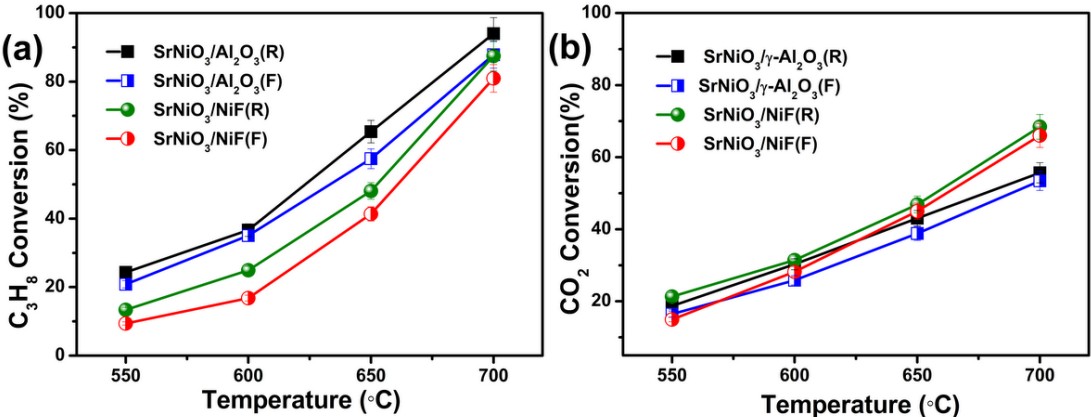

**Figure 6.** Catalytic activity of dry reforming of propane (DRP) as a function of temperature over $SrNiO_3/\gamma-Al_2O_3$ (F and R) and $SrNiO_3/NiF$ (F and R) catalysts at 550 °C to 700 °C under atmospheric pressure in CPR = 3 (30:10:60 $CO_2$:$C_3H_8$:Ar) and total flow rate = 200 mL/min (**a**) $C_3H_8$ conversion (**b**) $CO_2$ conversion.

The catalytic activity of bare supports is shown in Figure 7. As seen in Figure 7, both bare supports showed low catalytic activities even at 700 °C compared to $SrNiO_3$ loaded catalysts. The conversion of $C_3H_8$ of bare $\gamma-Al_2O_3$ shows slightly higher conversion at low temperature than bare NiF and the $C_3H_8$ conversion is almost the same (16%) at a higher temperature in both supports. The supports majorly contribute the $C_3H_8$ cracking reaction instead of DRP. Moreover, the $CO_2$ conversion of both supports shows the poor activity, which did not reduce to CO. The $CO_2$ conversion of $\gamma-Al_2O_3$ support showed the higher conversion (5%) at low temperatures than $C_3H_8$ conversion due to carbon formation from the $C_3H_8$ cracking reaction. Hence, the $SrNiO_3$ loaded catalysts show excellent activity compared to the bare supports, which indicates that the supports improve the dispersivity of active catalyst to enhance the interaction of the reactants with active sites.

The outlet concentration of $H_2$ and CO of the catalysts is shown in Figure 8. As seen in Figure 8, the $H_2$ production increases with increasing temperature for all catalyst. The $SrNiO_3/\gamma-Al_2O_3$ (F and R) catalysts showed that the higher $H_2$ production compares with $SrNiO_3/NiF$ (F and R) catalyst. The maximum $H_2$ and CO production were observed by $SrNiO_3/\gamma-Al_2O_3$ (R) is 22% and 20% at 700 °C respectively. It suggested that the $\gamma-Al_2O_3$ supported catalyst primarily enhanced the $C_3H_8$ cracking reaction instead of DRP, which leads to the coke generation. Notably, the $SrNiO_3/NiF$ (F and R)

catalysts have shown 20%, 21%, and 27%, 29% of $H_2$ and CO production, respectively. The $CO_2$ conversion always showed lower than the $C_3H_8$ conversion, and the $H_2$/CO ratio was significantly 0.7 for the $SrNiO_3$/NiF (F and R). This observation could be explained by a low amount of coke formation on the surface of the catalyst, and the DRP reaction took place in thermodynamic equilibrium.

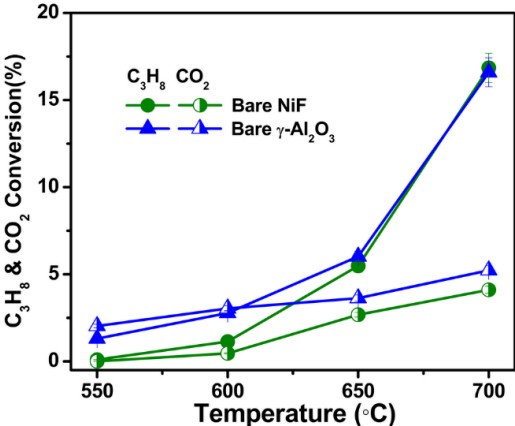

**Figure 7.** Catalytic activity of DRP as a function of temperature over bare supports $\gamma$-$Al_2O_3$ and NiF at 550 °C to 700 °C under atmospheric pressure in CPR = 3 (30:10:60 $CO_2$:$C_3H_8$:Ar), and total flow rate = 200 mL/min, $C_3H_8$ and $CO_2$ conversion.

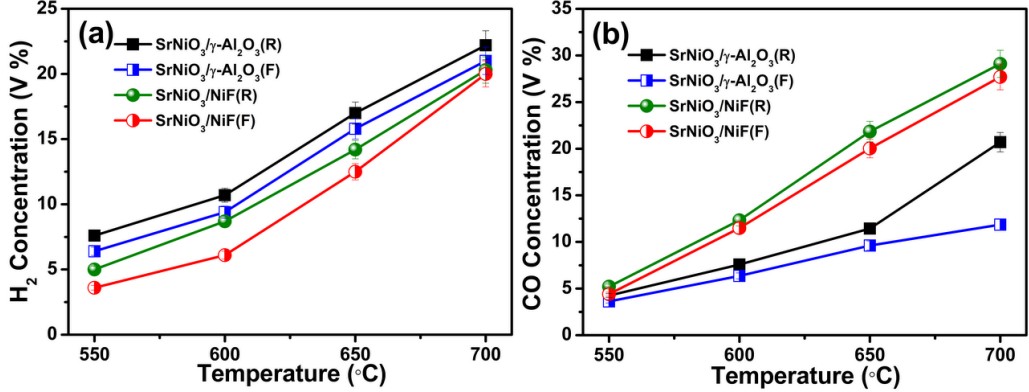

**Figure 8.** The product of DRP as a function of temperature over $SrNiO_3$/$\gamma$-$Al_2O_3$ (F and R) and $SrNiO_3$/NiF (F and R) catalysts at 550 °C to 700 °C under atmospheric pressure in CPR = 3 (30:10:60 $CO_2$:$C_3H_8$:Ar) and total flow rate = 200 mL/min (**a**) $H_2$ concentration (vol.%) and (**b**) CO concentration (vol.%).

As shown in Figure 9, we could compare the $SrNiO_3$ activity in two different supports materials at the isothermal condition. From the results, the NiF foam-supported catalyst showed superior catalytic activity. It could be active in two forms, and the results were almost the same. These suggested that the NiF could be facilitated the significant interaction of the catalytic surface and efficiently enhance the syngas production, which closes to the stoichiometric reaction. As a result of Figure 9, the $SrNiO_3$/NiF(R) revealed the significant conversion of $C_3H_8$ and $CO_2$ among all. Regarding CO and $H_2$ selectivity were 96% and 64%, respectively. Moreover, the $H_2$/CO ratio of the catalyst revealed that reaction mechanism, the $SrNiO_3$/$\gamma$-$Al_2O_3$(F and R) catalysts showed higher values 1.8 and 1.0, respectively, which suggested that the CO selectivity was less than the $H_2$ selectivity. The $CO_2$ conversion was lower than $C_3H_8$ conversion due to the additional $CO_2$ produced by water gas shift reaction (WGSR) (Equation (4)). The WGSR was confirmed by the value of $H_2$/CO ratio higher than 1 [34]. The CO selectivity severely affected due to the predominantly produced CO molecules could react with $H_2O$ to produce $CO_2$ and $H_2$. Notably, the $SrNiO_3$/NiF (F and R) catalysts have

shown the significant conversion of $C_3H_8$, $CO_2$ and higher selectivity of CO than the selectivity of $H_2$. The $H_2$/CO ratio is 0.7, which closes to the stoichiometric reaction value of DRP (Equation (2)). These indicated that the NiF supported catalysts performed excellently in terms of activity, which aid the DRP reaction and also significantly favor the side reaction reverse water gas shift reaction (Equation (5)). The $H_2$/CO ratio would be higher than 1 since this reaction did not occur [47,48].

$$CO + H_2O \rightarrow CO_2 + H_2 \tag{4}$$

$$CO_2 + H_2 \rightarrow H_2O + CO \tag{5}$$

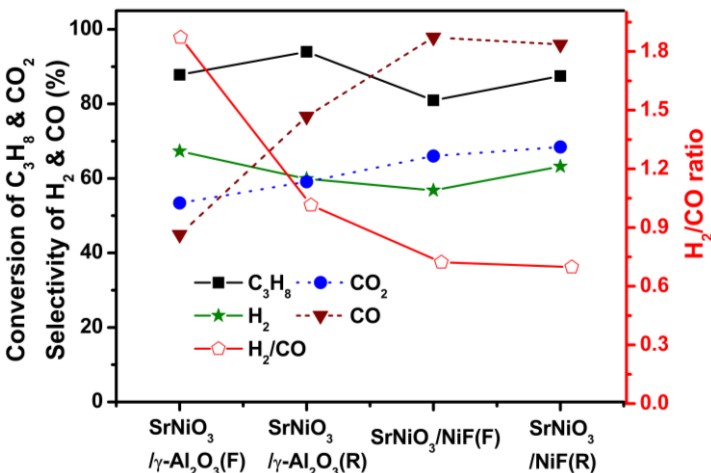

**Figure 9.** Catalytic performance of DRP over $SrNiO_3/\gamma$-$Al_2O_3$ (F and R) and $SrNiO_3$/NiF (F and R) catalysts at 700 °C, CPR = 3 (30:10:60 $CO_2$:$C_3H_8$:Ar) and total flow rate = 200 mL/min.

The catalyst stability of the $SrNiO_3$ perovskite catalysts was examined throughout 50 h at 700 °C shown in Figure 10. It indicated that the $SrNiO_3$/NiF(R) catalyst shows significantly stable catalytic activity during the DRP, among other catalysts. For instance, the $SrNiO_3$/NiF(R) showed no significant conversion loss of $C_3H_8$ and $CO_2$ was 85% and 66% from the begin 87% and 69%, respectively, after 50 h of the DRP. The significant activity and long-term stability of $SrNiO_3$/NiF(R) are exhibited for its high thermal stability, high dispersion, and gas permeability of support [30]. In notably, the unreduced $SrNiO_3$ perovskite catalyst showed a gradual decrease the catalytic activity even in two support materials for the period. The conversion of $C_3H_8$ and $CO_2$ over the $SrNiO_3/\gamma$-$Al_2O_3$(F) was 87% and 53% until 2 h of the reaction, which gradually decreased to 82% and 48%, respectively, after 50 h due to the amphoteric property of $\gamma$-$Al_2O_3$.

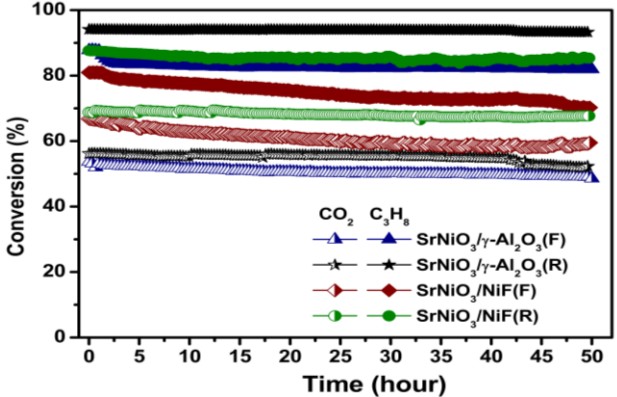

**Figure 10.** Stability of DRP over $SrNiO_3/\gamma$-$Al_2O_3$(F and R) and $SrNiO_3$/NiF (F and R) catalysts at 700 °C, CPR = 3 (30:10:60 $CO_2$:$C_3H_8$:Ar) and total flow rate = 200 mL/min for 50 h.

The catalytic stability results are shown in Table 2, the conversion of $C_3H_8$, $CO_2$ and the $H_2/CO$ ratio in the steady state at 700 °C after 50 h. The $C_3H_8$ and $CO_2$ conversion over $SrNiO_3/NiF(R)$ were superior to other catalysts. Besides, the $H_2/CO$ ratio was 0.7 for $SrNiO_3/NiF(R)$ and (F), which is closed to the stoichiometric value of DRP. Moreover, the results claimed that the higher selectivity of nickel foam-supported catalysts resist the carbon formation compared to $SrNiO_3/\gamma$-$Al_2O_3$(F) and (R). According to the results, The $SrNiO_3$ perovskite catalyst is a basic oxide. Due to its strong basicity, low surface area, and supported by NiF, $CO_2$ might interact actively with basic sites favoring the formation of carbonates to coverts CO, which minimize the carbon formation during DRP. Several authors [49–51] found that the highly basic catalysts like lanthanum oxide and $SmCoO_3$ are almost present in the carbonate phase on the catalyst surface after reforming reaction. These prevent the sintering of particles and extraction of particles from the surface of carbon filaments during the reaction.

The performance of the dry reforming of methane and propane over the perovskite-type catalysts were compared with the reported results, and the data are presented in Table 3. Strontium substituted perovskite catalyst were performed good catalytic conversion in dry reforming reaction and the $H_2/CO$ ratio of all the catalysts was less than the 1, which lower than the stoichiometry value of reforming reaction. The comparison suggested the excellent catalytic activity of $SrNiO_3$ supported catalysts compared to the reported materials.

**Table 2.** $C_3H_8$ and CO conversion, CO, $H_2$ selectivity, and $H_2/CO$ ratio.

| Catalyst | $X_{C_3H_8}$ (%) | $X_{CO_2}$ (%) | $S_{CO}$ (%) | $S_{H_2}$ (%) | $H/CO_2$ |
|---|---|---|---|---|---|
| $SrNiO_3/\gamma$-$Al_2O_3$ (F) | 82.0 | 48.0 | 20.2 | 61.0 | 1.9 |
| $SrNiO_3/\gamma$-$Al_2O_3$ (R) | 92.0 | 52.1 | 46.2 | 63.2 | 1.0 |
| $SrNiO_3/NiF$(F) | 70.2 | 59.5 | 94.1 | 64.3 | 0.7 |
| $SrNiO_3/NiF$(R) | 85.3 | 67.0 | 97.7 | 67.7 | 0.7 |

W = 1.0 g, P = 1 atm, T = 700 °C, TOS = 50 h.

**Table 3.** Performance of the perovskite catalysts in dry reforming of methane.

| Composition of Catalysts | Catalyst Preparation | Reforming Conditions | Conversion at 700 °C (%) | $H_2/CO$ Ratio | References |
|---|---|---|---|---|---|
| $LaNiO_3$ | Sol-gel | 700 °C, $CO_2$:$CH_4$ = 1 | 65 ($CH_4$) | 0.82 | [20] |
| Ni–Re/SS 316 | Electrodeposition | 700–800 °C, $CO_2$:$CH_4$ = 1 | 60 ($CH_4$) | 0.8 | [52] |
| $La_{0.8}Sr_{0.2}NiO_3$ | Sol-gel | 600–800°C, $CO_2$:$CH_4$ = 1 | 80.1 ($CH_4$) | NA | [2] |
| $La_{1-x}Sr_xNi_{0.4}Co_{0.6}O_3$ | Sol-gel | 700 °C, $CO_2$:$CH_4$ = 1 | 75 ($CH_4$) | 1.0 | [27] |
| $La_{0.9}Sr_{0.1}Ni_{0.5}Fe_{0.5}O_3$ | Sol-gel | 700–900 °C, $CO_2$:$CH_4$ = 1 | 47.2 ($CH_4$) | 0.7 | [53] |
| $Sr_{0.92}Y_{0.08}TiO_3$-3%wt Ni | Sol-gel | 625–745 °C, $CO_2$:$CH_4$ = 1 | 60 ($CH_4$) | NA | [54] |
| $SrTi_{0.85}Ru_{0.15}O_{3-\delta}$ | Sol-gel | 600–800 °C, $CO_2$:$CH_4$ = 1 | 75 ($CH_4$) | 0.8 | [55] |
| $SrNiO_3/NiF$(R) | Sol-gel | 550–700 °C, $CO_2$:$C_3H_8$ = 3 | 85.3 ($C_3H_8$) | 0.7 | This work |

### 2.3. Catalytic Characterization of Spent Catalyst

The post characterization of catalyst was examined by several techniques such as FE-SEM, Raman, temperature-programmed oxidation (TPO), and XRD to understand the catalyst after DRP at elevated temperature. Figure 11 showed the FE-SEM of all examined catalysts after DRP over time on stream (TOS) 50 h at 1µm magnification. As seen in Figure 11, the morphology of carbon formation on the catalyst was observed whisker/filament form. It is agreed with many previous reports because most of Ni present catalysts are the response to the carbon growth like tubular at high temperature [56,57].

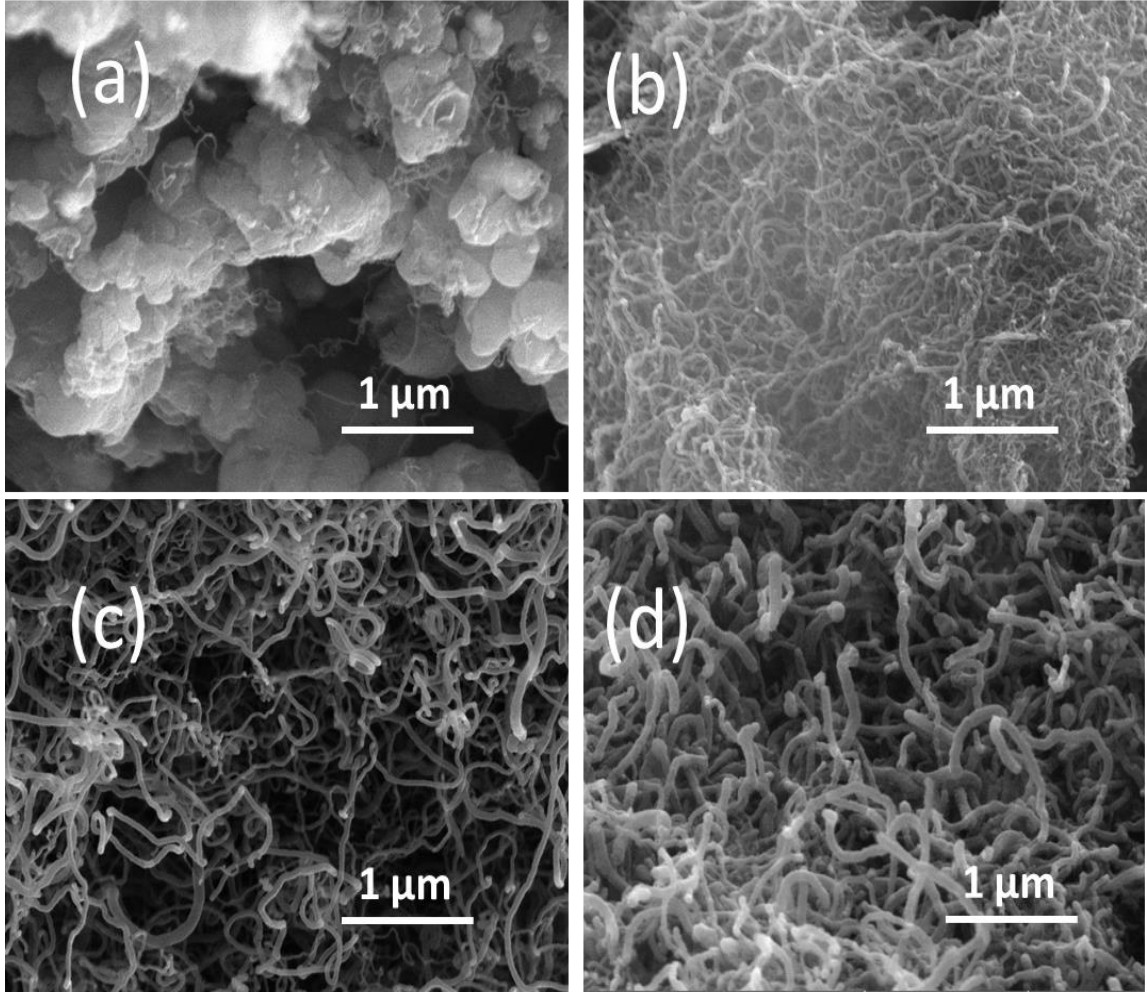

**Figure 11.** FE-SEM images of spent catalysts after DRP over 50 h (**a**) SrNiO$_3$/NiF(R), (**b**) SrNiO$_3$/NiF(F), (**c**) SrNiO$_3$/$\gamma$-Al$_2$O$_3$ (R), and (**d**) SrNiO$_3$/$\gamma$-Al$_2$O$_3$ (F).

As shown in Figure 11, the SrNiO$_3$/NiF (R) catalyst showed significantly low carbon on the surface among others, and the form of carbon also was a filament in nature. Figure 12 exhibited the carbon species of the all catalysts after DRP by Raman spectra. From the results, the carbon species were majorly in the form of graphitic. Three active peaks were observed in all catalysts. The first peak at 1337–1342 cm$^{-1}$ belongs to the D-band of Raman active mode of C—C bond stretching. The second peak at 1572–1580 cm$^{-1}$ corresponds to the G-band, which attributed to graphitized carbon form and also the third peak at 2678–2691 cm$^{-1}$ is attributed to the 2D-band of carbon nanotubes or filaments. The intensity of D-band (ID) and intensity of G-band (IG) could explain the graphitic disorder [58,59]. In this case of SrNiO$_3$/$\gamma$-Al$_2$O$_3$ (F and R) catalyst, the ID was always higher than the IG, which indicates deposited carbon could be in the form of graphite. The SrNiO$_3$/NiF (F and R) catalyst showed that the different trend of ID and IG was almost same, which suggested the existence of crystalline graphite [60].

TPO studies were done to understand the carbon species formation during DRP and its shown in Figure 13. The three primary types of carbon occurred on the surface of the catalyst after DRP. These types were distinguished by oxidation temperature of solid carbon with O$_2$. These are polymeric amorphous films or filaments (C$_\beta$) at 523–773 K, the vermicular carbon filaments/fibers/tubes (C$_\nu$) at 573–1273 K and the crystalline graphite (C$_C$) at 773–823 K [61–63].

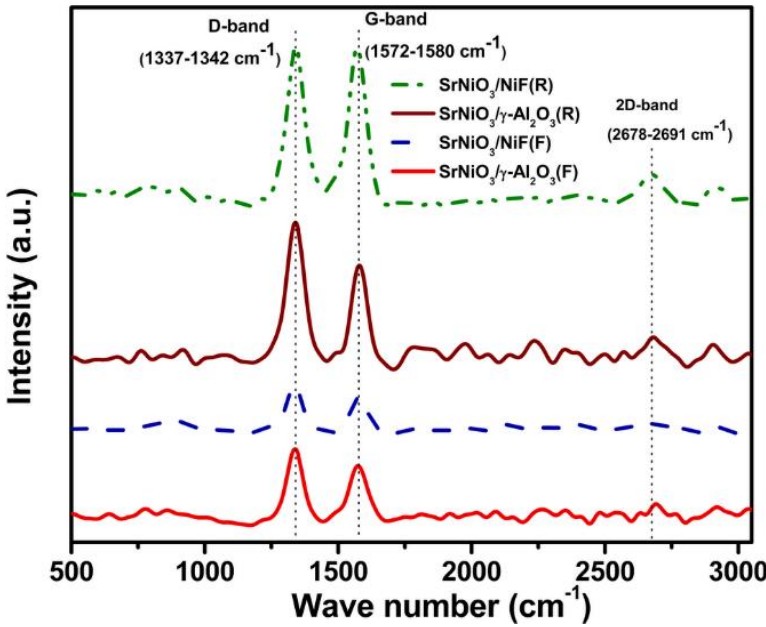

**Figure 12.** Raman spectra of spent catalysts after DRP over 50 h.

The results of the TPO profile, the reduced catalysts showed the three kinds of $C_\beta$, $C_C$, and $C_\nu$ on the surface, interestingly the both of reduced catalysts have vermicular carbon filaments. The un-reduce catalyst showed the polymeric amorphous films or filaments and crystalline graphite majorly [64]. These suggested that all catalyst produce carbon during DRP, even though the fresh catalyst showed a higher amount of carbon than reduced catalyst. The $\gamma$-Al$_2$O$_3$ supported catalysts produced more carbon instead of NiF, which suggested that the NiF allows increasing the molecule interaction due to their porous properties.

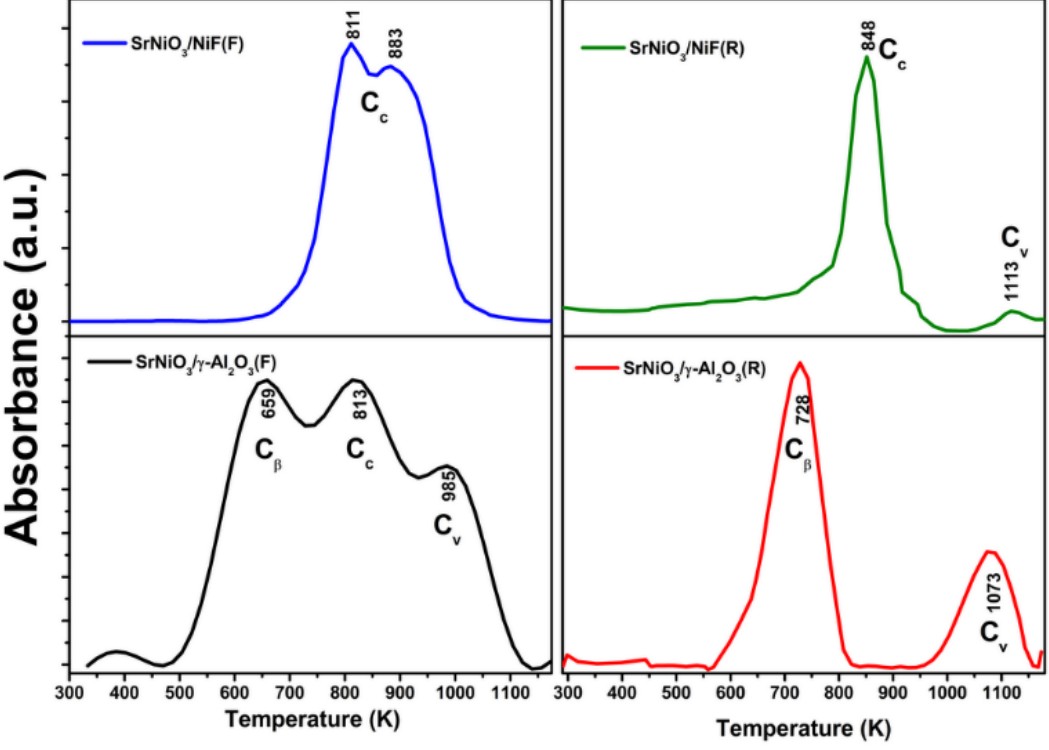

**Figure 13.** Temperature-programmed oxidation (TPO) profiles of used catalysts after DRP over 50 h.

XRD analysis of the used catalysts was examined after DRP stability test is shown in Figure 14. The patterns show most of the diffraction peaks were matched with the fresh catalysts patterns (Figure 1b,c). The patterns show the new three peaks at 25.8°, 46.5°, and 49.9° were identified in all catalysts, which corresponds to the $SrCO_3$ phase (JCPDS card no. 01-0556) [65]. Sutthiumporn et al. [56] reported that the Sr-doped $La_2O_3$ catalyst produces bidentate carbonate during the DRM reaction, which majorly reduces the carbon formation by the interaction of $CO_2$. The peak at a 2θ value of 25.8° corresponds to the carbon species of all catalysts, which overlapped with $SrCO_3$ peak. The reduced catalysts show the high intense peak of $SrCO_3$ due to SrO reacts with $CO_2$ to form $SrCO_3$ [66]. This result confirmed that the higher $CO_2$ conversion of reduced catalysts than unreduced catalysts.

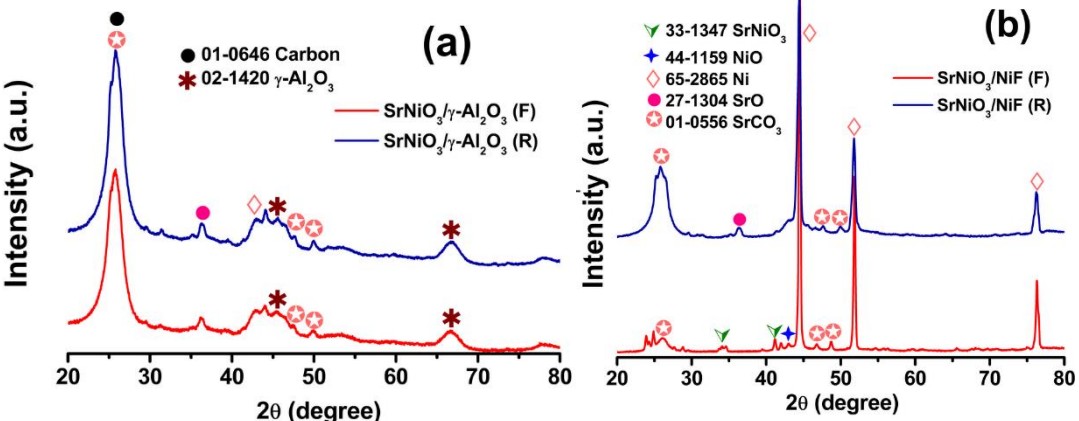

**Figure 14.** X-ray diffractogram of used catalyst (**a**) $SrNiO_3/\gamma\text{-}Al_2O_3$ (F and R) and (**b**) $SrNiO_3/NiF$ (F and R) after DRP for 50 h.

## 3. Experimental Section

### 3.1. Materials and Methods

Strontium nitrate ($Sr(NO_3)_2$, Nickel nitrate hexahydrate ($Ni(NO_3)_2 \cdot 6H_2O$), citric acid anhydrous, hydrochloric acid (HCl), and anhydrous ethanol ($C_2H_5OH$) were purchased from Daejung Chemicals Ltd., (Gyeonggi-do, South Korea). Gamma alumina ($\gamma\text{-}Al_2O_3$) pellets were purchased from Alfa Aesar Co., Inc, (Seoul, South Korea). Nickel foam (NiF) was purchased from MTI Crop., Ltd., (Seoul, South Korea). All the chemicals used in this experiment were of research grade, and de-ionized water (DI) was used in whole experiments.

### 3.2. Synthesis of $SrNiO_3$ Compound

$SrNiO_3$ based perovskite catalyst was prepared by the citric acid sol-gel method. In a typical preparation method, the precursor of 0.02 M of $Sr(NO_3)_2$, and $Ni(NO_3)_2 \cdot 6H_2O$ was dissolved in DI water in two separate beakers. These two solutions were mixed until to get a clear solution. Citric acid (0.06 m) in water was added by dropwise in the above nitrate solution to ensure miscibility. The solution was kept at 80 °C hot plate and stirred gently with Teflon-coated magnetic stir bar until the solution becomes viscous liquid. The resulting solution was kept at 110 °C for 12 h to dried off the excess of water. The dried powder was treated at 450 °C for 5 h to allow combustion reaction the resulting product was foamy. The final powder was ground well then calcined at 900 °C for ten hours, resulting in the synthesis of $SrNiO_3$ perovskite compound.

### 3.3. Preparation of Catalyst

The 10 wt.% of catalyst was prepared with different support materials ($\gamma\text{-}Al_2O_3$ and NiF) as follow procedure. In order to remove undesired materials on the surface of support materials, both supports were pretreated to remove the moisture in $\gamma\text{-}Al_2O_3$ and to remove oxide layers on NiF. The $\gamma\text{-}Al_2O_3$

pellets were washed several times with DI water to swipe out of loss bound particles on the surface final rinse with ethanol and then kept at 150 °C for 2 h to remove moistures. The purchased NiF was soaked in 1.0 M HCl solution for 30 min to remove the surface oxide layer and then washed several times with DI water to remove excess of acid and finally rinsed with ethanol and kept 110 °C for 1 h. The 16.0 mm diameter disc of pretreated NiF was cut by hole puncher to attain a disc shape. The desired amount (0.2 gm) of $SrNiO_3$ was ground with polyvinylidene difluoride (catalyst:PVDF ratio: 95:5) using N-methyl pyrrolidone (NMP) as a solvent to form a slurry. The 1.8 gm of the NiF discs (16 mm; OD) support was then coated with the prepared slurry, and the coated support was allowed to dry in an air oven at 85 °C for 12 h. The pretreated $\gamma$-$Al_2O_3$ pellets were ground well and made a fine powder, the 1.8 gm of fine $\gamma$-$Al_2O_3$ powder was mixed with 0.2 gm of $SrNiO_3$ powder to ensure the blending between $\gamma$-$Al_2O_3$ and $SrNiO_3$. A hydraulic press was used to prepare the pressured pellet, and the pellet was broken down into a small size by manually and sieved with 2 mm of mesh to remove the fine powder.

### 3.4. Material Characterization

X-ray diffractograms (XRD) were recorded an X-ray diffractometer (D/MAX 2200H, Bede 200, Rigaku Instruments C, Tokyo, Japan) with a horizontal goniometer performed by a fine focus copper X-ray tube (40 kV, 40 mA). BET specific surface area was measured by an Autosorb-1-MP instrument (Boynton Beach, FL, USA) at liquid nitrogen temperature. Prior to the analysis, the sample was degassed at 150 °C for 3 h, and the BET multi-point method was applied to estimate the surface area. The surface morphology of the synthesized materials was analyzed by Field-emission scanning electron microscopy (FE-SEM, JSM-6700F, JEOL, Tokyo, Japan). The Raman spectroscopy was evaluated on the samples using Raman HR Evolution Raman Spectrometer (LabRAM Horiba, Longjumeau, France) used at $Ar^+$ ion laser operating at 10 mW and a wavelength of 514 nm. Temperature-programmed desorption/reduction (TPD/TPR) experiments were carried out on the gas-chromatography (DS science, DS-6500, Gyeonggi-Do, South Korea) equipped with TCD detector. In case of TPR, the 100 mg of the catalyst sample was loaded in a U-shaped quartz tube and placed on the quartz wool. The temperature of the sample was measured using a thermocouple fixed with quartz tube near the sample. The sample was degassed at 250 °C for 30 min in a flow of Ar (45 mL/min) and to remove the moisture and other adsorbed gases from surface and pores of the sample and then the reactor cooled down to room temperature. The hydrogen was measured with the ramping of the sample temperature to 900 °C with a heating rate of 5 °C/min. Hydrogen consumption was measured by the difference in the thermal conductivity of the gas mixture., Aforementioned in TPR analysis until the degassed was same as the TPD analysis. Carry out $H_2$ adsorption at room temperature for 30 min by passing 5%$H_2$/Ar mixed gas with a flow rate of 50 mL/min. $H_2$ gas cut off and purge with pure Ar (45 mL/min) at room temperature for 30 min to remove physically adsorbed $H_2$. The $H_2$ desorption signal (as a function of time) was recorded by using a GC-TCD with a linear temperature increase from 25 °C to 800 °C with a heating rate of 5°C/min under Ar flow (45 mL/min). The amount of desorbed $H_2$ (mol) was measured the area under the curve. The following equation was used to calculate the amount of chemisorped $H_2$.

$$Area(mol) = \int_0^t [H_2]Q_{total}dt \tag{6}$$

where $[H_2]$ is a concentration of $H_2$ in mol/L and $Q_{total}$ is total flow rate L/min. Temperature-programmed oxidation (TPO) experiments of spent catalyst were carried out on the Fourier transform infrared spectroscopy (FTIR) (FTIR-7600 spectrometer, Lambda, Edwardstown, Australia) to understand the nature of carbon species. The 100 mg of spent catalyst was placed in a U-shaped quartz tube and flush with 50 mL/min of $N_2$ at 250 °C for 30 min prior to remove surface oxygen and to attain inert atmosphere and reactor cool down to room temperature with continues flow of $N_2$ gas. The 10%$O_2$/$N_2$ (50 mL/min) of oxidant gas feed was changed over, and outlet of gas

($CO_2$) was monitor through online FTIR at a ramped temperature from 25 °C to 900 °C at the rate of 5 °C/min.

### 3.5. Catalytic Activity and Selectivity

The catalytic activity of prepared catalysts was measured in fixed-bed quartz reactor (16 mm ID and 600 mm length). A 2.0 g of the catalyst was placed in the reactor between a sandwich of quartz wool. A tubular furnace has maintained the reaction temperature of the catalyst with an external temperature controller equipped with a K-type thermocouple. The reaction temperature of the catalyst was measured by a K-type thermocouple which fixed on it. The ratio of feed gases of carbon dioxide ($CO_2$) and propane ($C_3H_8$) (CPR) was 3, the total composition of feed gases are $C_3H_8$:$CO_2$:Ar in percentage (10:30:60) and controlled by a mass flow controller (MFC-500, Atovac Co., Yongin, Korea). The total flow rate of the reactant gas is 200 mL/min for all catalytic studies. The concentration of $C_3H_8$ and $CO_2$ were measured by online gas chromatography (GC- Micro-GCCP-4900, 10m PPQ column, Palo Alto, CA, USA) and the concentration of $H_2$ and CO were measured by GC (DS-Science, 20m- HayeSep-Q column, Gyeonggi-Do, South Korea) equipped with a thermal conductive detector (TCD). The catalytic activity of all catalysts was examined at every 50 °C interval of the temperatures from 550 °C to 700 °C. The conversion of reactants ($X_A$) and selectivity (S) of various products were calculated by the following equations [67].

$$X_A(\%) = \frac{C_{A_0} - C_A}{C_{A_0} + \varepsilon_A C_A} \times 100 \tag{7}$$

$$S_{H_2} = \frac{[H_2]_{out} \cdot F_{out}\left(\frac{mL}{min}\right)}{4 \cdot \left([C_3H_8]_{in} \cdot F_{in} - [C_3H_8]_{out} \cdot F_{out}\right)\left(\frac{mL}{min}\right)} \tag{8}$$

$$S_{CO} = \frac{[CO]_{out} \cdot F_{out}\left(\frac{mL}{min}\right)}{3 \cdot \left([C_3H_8]_{in} \cdot F_{in} - [C_3H_8]_{out} \cdot F_{out}\right) + \left([CO_2]_{in} \cdot F_{in} - [CO_2]_{out} \cdot F_{out}\right)\left(\frac{mL}{min}\right)} \tag{9}$$

$$\frac{H_2}{CO} = \frac{F_{H_2}}{F_{CO}} \tag{10}$$

where, $X_A$ is the conversion of $C_3H_8$ and $CO_2$, $C_{A0}$, and $C_A$ are inlet and outlet concentration of reactants $C_3H_8$ and $CO_2$, respectively, $F_{in}$ and $F_{out}$ are inlet and outlet flow rates (mL/min). $[C_3H_8]_{in}$, $[CO_2]_{in}$, $[C_3H_8]_{out}$, $[CO_2]_{out}$, $[H_2]_{out}$, and $[CO]_{out}$ were inlet and outlet concentration of $C_3H_8$, $CO_2$, $H_2$, and CO, respectively. The $F_{H2}$ and $F_{CO}$ molar flow rate of $H_2$ and CO in the outlet.

### 4. Conclusions

$SrNiO_3$ perovskite catalyst successfully synthesized by the citrate sol-gel method and used as a catalyst with $\gamma$-$Al_2O_3$ and NiF supports for the first time in propane dry reforming and found to have propane and $CO_2$ activity for syngas production efficiently. The effect of reduced $SrNiO_3$ perovskite catalyst on DRP has been investigated. For comparison study, the reduced and unreduced $SrNiO_3$ catalysts showed significant improvement in conversion of $C_3H_8$ and $CO_2$. The $SrNiO_3$/NiF(R) showed excellent activity regarding syngas production, the syngas produced with a significant selectivity of $H_2$ and CO and $H_2$/CO ratio was maintained close to the stoichiometric value. These measurements of catalytic activity with two different support materials have significantly affected the production of syngas. The results comprise to support perovskites on porous material like metallic foam and high surface area material like $\gamma$-$Al_2O_3$ to increase the number of exposed perovskite active sites. The strong basicity of strontium metal with NiF could aid the CO production and reduce carbon formation. The finding of this study has presented $SrNiO_3$ perovskite as a suitable catalyst in the DRP experiment

and also could be the replacement of rare earth perovskite in reforming reaction and cost effective with the less health-hazardous catalyst.

**Author Contributions:** S.M.S.P., M.M.H., G.G., performed the experimental work and analyzed the data; S.M.S.P. wrote the paper; Y.S.M. examined the experimental data, manuscript and supervised all the study.

**Funding:** This research received no external funding.

**Acknowledgments:** This work was supported by the Basic Science Research Program through the National Research Foundation funded by the Ministry of Science, ICT and Future Planning, Korea (Grant Nos. 2016R1A2A2A05920703 and 2018R1A4A1025998).

**Conflicts of Interest:** The authors declare no conflict of interest.

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
