# Peer review of "Dry Reforming of Propane over γ-Al2O3 and Nickel Foam Supported Novel SrNiO3 Perovskite Catalyst"

_catalysts, doi:10.3390/catal9010068_

Round 1

Reviewer 1 Report

The paper on “Dry reforming of Propane over gamma alumina and nickel foam supported novel SrNiO3 Perovskite catalyst” might be considered for publication after major revision due to the following points:

The resolution of the figures must be enhanced and the symbols should be enlarged as it is barely to differentiate between the samples to follow what is written in the text

For Fig.1 where is the Ni0 on the figure and how about the XRD of the supported perovskite. It must be added

The perovskite has no mesopores the sample BET is only 3.3 m2/g and from the isotherm no hysteresis

How the active site was calculated to be 3.79 x 1020 a reference and the method must be added in details. Second what about the TPR of the supported perovskite these figures must be added. In literature 35 the author used H2-TPD to calculate the active site and nothing about spill over of H2. Also the H2-TPD of the supported catalysts should be measured

Line 128 is it dry reforming of methane or propane?!!!!!!

Line 129 what does the author want to say with “reduction step on long run has been reported to improve the reactant conversion as well as syngas formation”.  This sentence must be clarified

Results of the perovskite without support must be added.In line 133 what is the increasing trend with increases temperature? What trend ? in line 136 the same trend was observed in the case of Ni supported catalysts what other Ni supported catalysts the author mean here . Please clarify.  

In all the catalytic figures the conditions of the reaction are missed and what is CPR?

In line 160 the formation of coke leads to sever deactivation but where is the deactivation the symbol are very small that it is not possible to differentiate between them

In line 191 The catalyst stability of the SrNiO3 pervoskite catalysts

In line 195 The significant activity and stability for SrNiO3/NiF(R) due to high thermal stability high dispersion microstructure of the support and so on where is the prove for all of this even the author mention about the basicity and no test to prove this !!!!!! please delete or prove

In the experimental part more details about how H2-TPD and TPR was done, TPO. Also it is written for TPR and TPD was carried on gas chromatography it is not really clear how to perform the measurement with GC

Comparing the results with literature should be added

English must be revised precisely because many grammatical mistakes that make it difficult to understand the text

Author Response

    We thank the reviewer for the providing your valuable time for the peer-review processing of our manuscript and insightful comments for our manuscript. A point by point response to your comments/suggestion is given in the attached file. 

Reviewer 2 Report

Please find attached my suggestions and comments

Author Response

We thank the reviewer for the providing your valuable time for the peer-review processing of our manuscript and insightful comments for our manuscript. A point by point response to your comments has been answered in the attached file.

Round 2

Reviewer 1 Report

Two more comments I think for the BET for such low surface area it is not possible to apply the BJH method as this method is applicable for the mesopores and the author what he measures is only the voids between the particles 

I think 10 wt.% of the loading is not less amount at all and probably it would better to increase the amount of the catalyst so it is possible to get better signal and indeed I am wondering how the GC can give continuously signal. With TCD it is possible

The law the author mentioned to calculate the active site is not correct one this law can be used for calculating the amount of H2 consumption but how he calculated the active site there is another reference J.R.Anderson, K.C. Pratt  „Introduction to Characterization and testing of catalysts Academic Press 1985 page 7

These notes should be changed and after that I think it is possible to publish

Author Response

    We thank the reviewer for evaluating our manuscript and giving the valuable opinion/suggestion to improve the manuscript. A point by point responses to your comments/suggestion is provided in the attached file.

Reviewer 2 Report

Authors responded well to the suggestions, I recommend the manuscript to be accepted.

Author Response

We are pleased to thank you for accepting our manuscript.